# Contrastive Latent Variable Models for Neural Text Generation

**Zhiyang Teng**[1,2]   **Chenhua Chen**[1,2]   **Yan Zhang**[3]   **Yue Zhang**[1,2]

[1]School of Engineering, Westlake University, China
[2]Institute of Advanced Technology, Westlake Institute for Advanced Study, China
[3]National University of Singapore

## Abstract

Deep latent variable models such as variational autoencoders and energy-based models are widely used for neural text generation. Most of them focus on matching the prior distribution with the posterior distribution of the latent variable for text reconstruction. In addition to instance-level reconstruction, this paper aims to integrate contrastive learning in the latent space, forcing the latent variables to learn high-level semantics by exploring inter-instance relationships. Experiments on various text generation benchmarks show the effectiveness of our proposed method. We also empirically show that our method can mitigate the posterior collapse issue for latent variable based text generation models.

## 1 INTRODUCTION

Deep latent variable models such as variational autoencoder (VAEs) [Bowman et al., 2015] and deep energy-based models [Deng et al., 2020, Pang et al., 2020] have been widely used for text generation applications, such as machine translation [Calixto et al., 2019], language modeling [Deng et al., 2020], dialogue understanding [Shen et al., 2017], and story generation [Chen et al., 2021, Jhamtani and Berg-Kirkpatrick, 2020]. These models follow a sequence-to-sequence [Sutskever et al., 2014] task setting, first stochastically mapping the input sentence into a latent variable according to proper probabilistic distribution assumptions and then reconstruct the whole sentence. By manipulating the latent variable, controllable text generation models with both diversity and fluency can be achieved. In such models, the latent variable is expected to encode high-level semantic and style information of the input sequence, which can serve to guide the generation of output sequences in different text generation tasks.

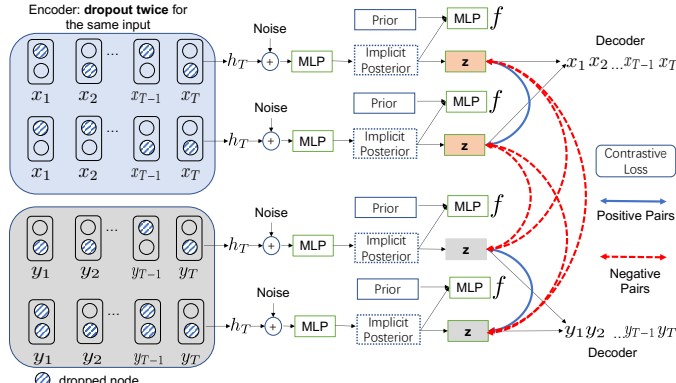

Figure 1: Contrastive learning over latent variables.

In this paper, we consider VAE and its variant iVAE [Fang et al., 2019], which addresses two limits of simple VAEs. The first is the simple posterior, which is usually assumed to be isotropic Gaussian, limiting the expressive power of the latent variable. Many efforts have been made to use an enhanced posterior [Casale et al., 2018, Tomczak and Welling, 2018]. Another problem is the posterior collapse issue [Bowman et al., 2015], where the learned latent variable can be largely ignored in a trained model. iVAE solves these problems by using implicit representation of posterior distributions of latent variables, where the posterior distribution are represented by a set of samples. Mutual information maximization is considered to encourage the correlation between the inputs and the induced latent variables.

It has been shown that the strength of iVAE as compared to vanilla VAE lies in better latent variable representations [Fang et al., 2019]. In particular, the latent variable should faithfully represent the semantics of the input, while allowing diversified outputs. Intuitively, further improvements towards these two characteristics in the latent variable can potentially lead to stronger VAE methods. However, there is an intrinsic tradeoff between diversity and faithfulness, as

*Accepted for the 38th Conference on Uncertainty in Artificial Intelligence* (UAI 2022).

it is more challenging to ensure that a set of different latent representations all embody the same semantic information. To this end, both VAE and iVAE make use of the reconstruction objective, learning to reconstruct the same input from different latent variable samples. The effectiveness can be limited by the number of samples that can be drawn, which can hardly match the number of possible variations in the reconstruction output. To address this issue, we consider adding direct supervision signals on the latent representation itself, by ensuring that different latent variable samples of the same input are closer in the vector space as compared to latent variable samples of two different sentences. This loss ensures semantic faithfulness without losing diversity. It can be viewed as belonging to the category of contrastive loss [Khosla et al., 2020, Le-Khac et al., 2020].

Our model structure is shown in Figure 1. In particular, for VAE, we encode the same input sentence twice with different dropout masks and then sample the corresponding latent codes for obtaining positive pairs. Negative pairs are constructed by comparing the sampled latent vector with the remaining latent vectors resided in the same batch. Contrastive learning is used to increase the semantic similarity of positive pairs and decrease the similarities of negative pairs. For iVAE, we observe that the KL divergence between the implicit posterior distribution and the prior distribution can explode. The main reason is that, to calculate the KL divergence between sample-based representations, iVAE uses an approximation based on Fechel duality theorem [Rockafellar et al., 1966, Dai et al., 2018]. However, this approximation cannot guarantee the non-negativity of KL divergence. We empirically observe that contrastive learning on the latent space can ease the explosion problem. We further adapt this approximation by borrowing the idea from CircleLoss [Sun et al., 2020] to ensure the non-negativity of the KL approximation.

We conduct experiments on a synthetic dataset and three language modeling benchmarks. Results show that our model can learn better latent representations on the synthetic dataset and achieve better perplexity scores than VAE and iVAE. For example, on PennTreebank language modeling benchmark, our model can decrease the perplexity scores by more than 10 points compared to iVAE. Including contrastive learning over latent variable in VAEs can alleviate the posterior collapse issues and avoid approximated KL divergence explosion in iVAEs. To our knowledge, we are the first to combine contrastive learning with latent variable models for natural language processing. We make our models and codes publicly available at `https://github.com/zeeeyang/constrastive_vae` and an alternative implementation using MindSpore[1] which is a new deep learning computing framework, can be found at `https://github.com/zeeeyang/constrastive_vae_mindspore`.

---

[1] `https://www.mindspore.cn/`

## 2 RELATED WORK

**Varational Auto-Encoders for Text Generation** VAE [Bowman et al., 2015] proposes the first model to enable text generation from continuous space using variational inference and an isotropic Gaussian prior. Many efforts have been made to improve VAE by using advanced prior distributions [Tomczak and Welling, 2018, Wang and Wang, 2019, Ding and Gimpel, 2021] and alleviate the the posterior collapse issue of VAE [Bowman et al., 2015, Higgins et al., 2017, He et al., 2019, Fu et al., 2019], making the generation depends on latent representations. iVAE [Fang et al., 2019] uses implicit sample-based representation, without requiring an explicit density form for the approximate posterior, which enables more flexibility. To solve the posterior collapse issue, iVAE adopts a mutual information regularization to match the aggregated posterior to the prior distribution. APO-VAE [Dai et al., 2018] defines both the prior and posterior of latent variables over a Poincaré ball in hyperbolic space, which also adopts the training scheme of iVAE. We do not compare with Apo-VAE since it additionally adopts a data-dependent VampPrior [Tomczak and Welling, 2018]. Our model is based on iVAE, with direct supervision of latent variables using contrastive learning.

**Contrastive Learning for Sentence Representations** Contrastive learning [Oord et al., 2018, Hjelm et al., 2019, He et al., 2020, Chen et al., 2020] has achieved great success in self-supervised visual representation learning. Recent work transfers this learning strategy to texts with different network architectures and augmenting methods for unsupervised sentence representation learning [Zhang et al., 2020, Giorgi et al., 2021, Carlsson et al., 2021, Gao et al., 2021]. Among these, SimCSE [Gao et al., 2021], which only uses standard dropout as minimal data augmentation, achieves the state of art and even performs on par with previously supervised counterparts. Our method uses the same data augmentation method as SimCSE, applying dropout to the input batch twice to obtain two different views. Similarly, R-drop [Liang et al., 2021] applies dropout twice to regularize the behaviours of the decoders in language modeling. However, different from SimCSE and R-drop, the goal of our method is to better supervise the *latent state*s of variational auto-encoders.

**Contrastive Learning for Text Generation** Dai et al. [2021] use CPC to do utterance-level contrastive learning between the dialogue context and the corresponding response. Su et al. [2022] propose a contrastive training objective for text generation, which is used to improve MLE based training and beam search decoding. In their methods, the hidden vector of a token is contrasted with the hidden representation vectors of the remaining tokens in the same sentence, which is different from our method. To our knowledge, we are the first to directly apply contrastive learning over the *latent* vectors for variational auto-encoders. Lee et al. [2021]

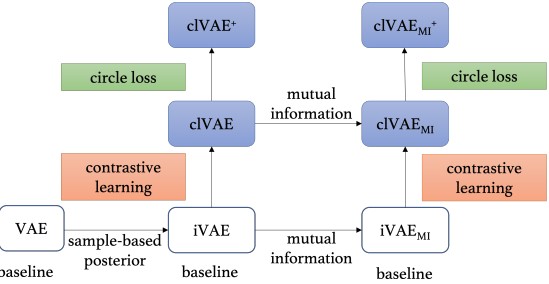

Figure 2: Relationships among the models.

combines adversarial perturbations with contrastive learning to solve the exposure bias problem for conditional sequence generation. Their contrastive learning is done in the embedding space, which is different from our method.

# 3 METHOD

Figure 2 shows the models we investigated in this paper. The blue boxes denote our proposed models. We introduce the VAE baselines in Sec 3.1. In Sec 3.2, we describe iVAE, which considers sample-based posterior distribution with VAE, and a variant of iVAE (iVAE$_{MI}$) which considers mutual information between input data and latent. In Sec 3.3, we first show two models clVAE and clVAE$_{MI}$ by integrating contrastive learning over latent variables for text generation based on iVAE and iVAE$_{MI}$, respectively. Then, we discuss clVAE$^+$ and clVAE$^+_{MI}$, which solve the explosion problem of approximated KL divergence by using circle loss enhancement.

## 3.1 VARIATIONAL AUTO-ENCODER BASELINE

Formally, given a sentence $\mathbf{x} = \{x_1, x_2, \ldots, x_T\}$, where $T$ is the length of $\mathbf{x}$, an auto-regressive language model finds a neural network $\boldsymbol{\theta}$ that can maximize the log-likelihood $\log P(\mathbf{x}; \boldsymbol{\theta}) = \prod_{i=1}^{n} \log P(x_i|x_1, \ldots, x_{i-1}; \boldsymbol{\theta})$. Variational auto-encoder (VAE) introduces a random latent variable $\boldsymbol{z}$ to model $P(\mathbf{x}; \boldsymbol{\theta}) = \int_{\boldsymbol{z}} P(\boldsymbol{z}; \boldsymbol{\theta}) P(\mathbf{x}|\boldsymbol{z}; \boldsymbol{\theta}) d\mathbf{z}$. To generate a sentence $\mathbf{x}$, a $\boldsymbol{z}$ is first sampled from the *prior distribution* $P(\boldsymbol{z}; \boldsymbol{\theta})$ and then a *decoder network* is used to produce $\mathbf{x}$ from $\boldsymbol{z}$ according to $P(\mathbf{x}|\boldsymbol{z}; \boldsymbol{\theta})$. In this paper, $\boldsymbol{z}$ is is continuous, $\log P(\mathbf{x}; \boldsymbol{\theta})$ becomes intractable due to the integration over $\boldsymbol{z}$. To train $\boldsymbol{\theta}$, a variational posterior distribution $Q(\boldsymbol{z}|\boldsymbol{x}; \boldsymbol{\phi})$ over $\boldsymbol{z}$ is introduced to approximate the true posterior $P(\boldsymbol{z}|\boldsymbol{x}; \boldsymbol{\theta})$. To ensure that $\boldsymbol{z}$ from the prior $P(\boldsymbol{z}; \boldsymbol{\theta})$ can encode meaningful semantic representations of the input sentence, $Q(\boldsymbol{z}|\mathbf{x}; \boldsymbol{\phi})$ is forced to match $P(\boldsymbol{z}; \boldsymbol{\theta})$ during training. Based on $Q(\boldsymbol{z}|\mathbf{x}; \boldsymbol{\phi})$, an evidence lower bound (ELBO) [Bowman et al., 2015] to $\log P(\mathbf{x}; \boldsymbol{\theta})$ is

defined as:

$$\text{ELBO}(\mathbf{x}; \boldsymbol{\phi}, \boldsymbol{\theta}) \tag{1}$$
$$= \underbrace{\mathbb{E}_{\boldsymbol{z} \sim Q(\boldsymbol{z}|\mathbf{x}; \boldsymbol{\phi})} \log P(\mathbf{x}|\boldsymbol{z}; \boldsymbol{\theta})}_{\text{reconstruction loss}} - \underbrace{\text{KL}\Big(Q(\boldsymbol{z}|\mathbf{x}; \boldsymbol{\phi}), P(\boldsymbol{z}; \boldsymbol{\theta})\Big)}_{\text{regularizer}}.$$

$\boldsymbol{\theta}$ and $\boldsymbol{\phi}$ are jointly trained by maximizing ELBO in Eq 1, where the first term is a reconstruction loss. The encoder first generates $\boldsymbol{z}$ according to $Q(\boldsymbol{z}|\mathbf{x}; \boldsymbol{\phi})$, which is called an *encoder network* or a *recognition network*. $\boldsymbol{z}$ is required to reconstruct $\mathbf{x}$ by maximizing $\log P(\mathbf{x}|\boldsymbol{z}; \boldsymbol{\theta})$. The second term of Eq 1 is the Kullback-Leibler divergence, which regularizes the posterior distribution to be close to the prior distribution. The posterior distribution and the prior distribution are often assumed to belong to the same parametric distributions with different parameters, which can ease the learning burden. Ideally, a good model maximizes the first term and minimizes the second term. When the KL term tends to be 0, $Q(\boldsymbol{z}|\mathbf{x}; \boldsymbol{\phi})$ degenerates to $P(\boldsymbol{z}; \boldsymbol{\theta})$ and the posterior collapse issue happens. For text generation, it is common due to the auto-regressive decoder can be very strong, which totally ignores $\boldsymbol{z}$ and generates $x_i$ solely from $[x_1, \ldots, x_{i-1}]$ [Fu et al., 2019].

## 3.2 IMPLICIT VAE BASELINE (IVAE)

$Q(\boldsymbol{z}|\boldsymbol{x}; \boldsymbol{\phi})$ is typically assumed to be multivariate Gaussian distributions for computational convenience. However, Gaussians are not enough to capture the rich semantics of natural sentences in latent space. Implicit VAEs (iVAE) introduce a sample-based posterior representation which does not depend on an explicit density form. $Q(\boldsymbol{z}|\boldsymbol{x}; \boldsymbol{\phi})$ is represented by a set of samples $\{\boldsymbol{z}_{\boldsymbol{x},i}\}_{i=1}^{S}$, where $S$ is the sample size. The $i$-th sample is given by

$$\boldsymbol{\xi}_i \sim \mathcal{N}(\boldsymbol{\xi}), \boldsymbol{z}_{\boldsymbol{x},i} = \text{ENC}(\boldsymbol{x}, \boldsymbol{\xi}_i; \boldsymbol{\phi}), \tag{2}$$

where $\mathcal{N}(\boldsymbol{\xi})$ is the standard Gaussian, $\text{ENC}(\boldsymbol{x}, \boldsymbol{\xi}_i; \boldsymbol{\phi})$ is a noise aware encoder network. The sentence representation $\mathbf{h}_T$ of $\boldsymbol{x}$ is generated by a LSTM encoder [Hochreiter and Schmidhuber, 1997] and $\mathbf{h}_T$ is concatenated with $\boldsymbol{\xi}_i$ by a multi-layer perceptron layer (MLP) to produce $\boldsymbol{z}_{\boldsymbol{x},i}$.

The KL divergence of Eq 1 is intractable after using the sample based posterior representation. Following Fang et al. [2019], its dual form is calculated according to Fenchel duality theorem [Rockafellar et al., 1966, Dai et al., 2018] by introducing an auxiliary function $f(\boldsymbol{x}, \boldsymbol{z}; \boldsymbol{\psi})$,

$$\text{KL}\left(Q(\boldsymbol{z}|\boldsymbol{x}; \boldsymbol{\phi}), P(\boldsymbol{z}; \boldsymbol{\theta})\right) \tag{3}$$
$$= \max_f \mathbb{E}_{\boldsymbol{z} \sim Q(\boldsymbol{z}|\boldsymbol{x}; \boldsymbol{\phi})} f(\boldsymbol{x}, \boldsymbol{z}; \boldsymbol{\psi}) - \mathbb{E}_{\boldsymbol{z} \sim P(\boldsymbol{z}; \boldsymbol{\theta})} \exp(f(\boldsymbol{x}, \boldsymbol{z}; \boldsymbol{\psi})),$$

where $f$ outputs a real value and $\boldsymbol{\psi}$ is the model parameters for $f$. $f$ is implemented as a MLP, which distinguishes between $(\boldsymbol{x}, \boldsymbol{z}_Q)$ and $(\boldsymbol{x}, \boldsymbol{z}_P)$ where $\boldsymbol{z}_Q$ and $\boldsymbol{z}_P$ denote latent

samples drawn from the posterior and the prior distributions, respectively. Using Eq 3, ELBO can be written as

$$\mathcal{L}_{\text{iVAE}} = \mathbb{E}_{z \sim Q(z|x;\phi)} \log P(x|z;\theta) - \mathbb{E}_{z \sim Q(z|x;\phi)} f(x, z, \psi)$$
$$+ \mathbb{E}_{z \sim P(z;\theta)} \exp(f(x, z, \psi)). \tag{4}$$

Considering a dataset $\mathbf{D} = \{x_i\}_{i=1}^n$, the loss function of the whole $\mathbf{D}$ using VAE and iVAE are:

$$\mathcal{L}_{\text{VAE}} = \mathbb{E}_{x \sim D}\Big[\mathbb{E}_{z \sim Q(z|x;\phi)} \log P(x|z;\theta)\Big]$$
$$- \mathbb{E}_{x \sim D}\Big[\text{KL}(Q(z|x;\phi), P(z;\theta))\Big], \quad (5)$$

$$\mathcal{L}_{\text{IVAE}} = \mathbb{E}_{x \sim D}\Big[\mathbb{E}_{z \sim Q(z|x,\phi)} \log P(x|z,\theta)\Big]$$
$$- \mathbb{E}_{x \sim D}\Big[\mathbb{E}_{z \sim Q(z|x,\phi)} f(x, z, \psi)$$
$$+ \mathbb{E}_{z \sim P(z;\theta)} \exp(f(x, z, \psi))\Big]. \quad (6)$$

**Mutual Information Regularized iVAE (iVAE$_{\text{MI}}$)** In Eq 1, the KL term forces a data-dependent posterior $Q(z|x, \phi)$ to match the same data-agnostic prior distribution $P(z; \theta)$. An variant of iVAE uses aggregated posterior $Q(z; \phi)$ to match $P(x|z; \theta)$, where $Q(z; \phi) = \int Q(x)Q(z|x; \phi)dx$ and $Q(x)$ is the empirical data distribution. Using aggregated posterior $Q(z; \phi)$, the latent space can be better regularized by cooperating different posterior distributions of all sentences to jointly match the prior. Given a dataset $D = \{x_i\}_{i=1}^n$, replacing the expected KL term $\mathbb{E}_{x \sim D}\text{KL}(Q(z|x; \phi), P(z; \theta))$ with $\text{KL}(Q(z; \phi), P(z; \theta))$, the new training objective is,

$$\mathcal{L}_{\text{IVAE}_{\text{MI}}} = \mathbb{E}_{x \sim D}\Big[\mathbb{E}_{z \sim Q(z|x;\phi)} \log P(x|z;\theta)\Big] \quad (7)$$
$$- \mathbb{E}_{z \sim Q(z;\phi)} g(z, \psi) + \mathbb{E}_{z \sim P(z;\theta)} \exp(g(z, \psi)),$$

where $g$ is a similar function as $f$ to produce a real value. The dual form of $\text{KL}(Q(z; \phi), P(z; \theta))$ is $\max_g \mathbb{E}_{z \sim Q(z;\phi)} g(z, \psi) - \mathbb{E}_{z \sim P(z;\theta)} \exp(g(z, \psi))$. Differently, $g$ only considers $z$ instead of the concatenation of $x$ and $z$. To generate samples from $Q(z, \phi)$, ancestral sampling can be used. First, $x$ is sampled from $D$ and then $z$ is sampled from $Q(z|x; \phi)$. Fang et al. [2019] show that optimizing the aggregated posterior based KL term is to maximize the mutual information $I(x, z)$ under the joint distribution $Q(x, z; \phi)$. We take the above model (**iVAE$_{\text{MI}}$**) as our main baseline.

### 3.3 CONTRASTIVE LEARNING OVER LATENT VARIABLES FOR TEXT GENERATION

As shown in Figure 1, the overall structure for our proposed model is based on the framework of iVAE. It contains three main components, an encoder, a contrastive learning module and a decoder. The encoder is quite similar to iVAE, except that dropout is enabled. The decoder is exactly the same as

iVAE. The contrastive learning module is our main design. In this section, we show how to integrate contrastive learning over latent variables with iVAE.

Formally, given the latent sample $z_i$ of the $i$-th input sentence, its positive latent sample $zp = \{z_i^+\}$ which is semantically similar to $z_i$, its $M$ negative latent samples $zn = \{z_{i,j}^-\}_{j=1}^M$ which is semantically far away from $z_i$, the training objective using contrastive learning for the $i$-th sentence is to maximize:

$$\ell_i = \log \frac{\exp(\text{SIM}(\mathbf{z}_i, \mathbf{z}_i^+)/\tau)}{\sum_{\mathbf{z} \in \{zp \cup zn\}} \exp(\text{SIM}(\mathbf{z}_i, \mathbf{z})/\tau)}, \quad (8)$$

where $\tau$ is a temperature hyper-parameter and $\text{SIM}(z_1, z_2)$ is a function for measuring semantic similarities between $z_1$ and $z_2$. Particularly, we use cosine similarity as the semantic similarity metric,

$$\text{SIM}(\mathbf{z}_1, \mathbf{z}_2) = \frac{\mathbf{z}_1^{\text{T}} \mathbf{z}_2}{\|\mathbf{z}_1\|_2 \cdot \|\mathbf{z}_2\|_2}, \quad (9)$$

where $\|\mathbf{z}\|_2$ denotes the $\mathcal{L}_2$ norm of $z$. Following Chen et al. [2020] and Gao et al. [2021], we consider batch-based contrastive learning. For each sentence $x_i$, we follow Gao et al. [2021] and use dropout as data augmentation. We encode the input sentence twice with dropout being enabled, and obtain two different views $x_i$ and $x_i^+$ of the same sentence. $x_i$ is then regarded as the anchor sentence and $x_i^+$ is the positive sample of $x_i$. The remaining sentences in the same batch are treated as the negative samples of $x_i$. In this way, the number of negative examples $M = N - 1$. To obtain the corresponding posterior latent variables, we sample $z_i \sim Q(z|x_i; \phi)$ and $z_i^+ \sim Q(z|x_i^+; \phi)$. For the posterior samples $z_j \sim Q(z|x_j; \phi)$ and $j \neq i$, they are the negative samples of $z_i$. Denote $\mathbf{u}_i = \{\mathbf{z}_i^+\} \cup \{\mathbf{z}_{i,j}^-\}_{j=1}^M$ and the overall contrastive learning loss for the whole batch is

$$\mathcal{L}_{\text{CL}} = \frac{1}{N} \sum_{i=1}^N \log \frac{\exp(\text{SIM}(\mathbf{z}_i, \mathbf{z}_i^+)/\tau)}{\sum_{\mathbf{z} \in \mathbf{u}_i} \exp(\text{SIM}(\mathbf{z}_i, \mathbf{z})/\tau)}. \quad (10)$$

Adding the contrastive loss into $\mathcal{L}_{\text{IVAE}}$ and $\mathcal{L}_{\text{IVAE}_{\text{MI}}}$, we have

$$\mathcal{L}_{\text{CLVAE}} = \mathcal{L}_{\text{IVAE}} + \mathcal{L}_{\text{CL}}, \quad (11)$$
$$\mathcal{L}_{\text{CLVAE}_{\text{MI}}} = \mathcal{L}_{\text{IVAE}_{\text{MI}}} + \mathcal{L}_{\text{CL}}. \quad (12)$$

**Improved Dual Function** $\text{KL}(Q(z|x; \phi), P(z; \theta))$ should be non-negative according to its definition. However, the approximation by the dual function in Eq 3 cannot ensure this property since $\mathbb{E}_{z \sim Q(z|x;\phi)} f(x, z; \psi)$ can be less than $\mathbb{E}_{z \sim P(z;\theta)} \exp(f(x, z; \psi))$. To encourage the KL approximation to be non-negative, we resort to a new constraint $\mathbb{E}_{z \sim Q(z|x;\phi)} f(x, z; \psi) > 0 > \mathbb{E}_{z \sim P(z;\theta)} \exp(f(x, z; \psi))$. Using this constraint, we define a new approximation inspired by circle loss [Sun et al., 2020],

$$\text{KL}(Q(z|x; \phi), P(z; \theta)) \quad (13)$$
$$= \min_f \log\Big(1 + \exp\big(-\mathbb{E}_{z \sim Q(z|x;\phi)} f(x, z; \psi)\big)\Big)$$
$$+ \log\Big(1 + \exp\big(\mathbb{E}_{z \sim P(z;\theta)} \exp(f(x, z; \psi))\big)\Big),$$

To minimize Eq 13, $\mathbb{E}_{\boldsymbol{z} \sim Q(\boldsymbol{z}|\boldsymbol{x};\boldsymbol{\phi})} f(\boldsymbol{x}, \boldsymbol{z}; \boldsymbol{\psi})$ should be forced to be much greater than 0 and $\mathbb{E}_{\boldsymbol{z} \sim P(\boldsymbol{z};\boldsymbol{\theta})} f(\boldsymbol{x}, \boldsymbol{z}; \boldsymbol{\psi})$ should be much less than 0. Similarly, $\mathsf{KL}(Q(\boldsymbol{z}; \boldsymbol{\phi}), P(\boldsymbol{z}; \boldsymbol{\theta}))$ can be approximated as,

$$\mathsf{KL}\left(Q(\boldsymbol{z}; \boldsymbol{\phi}), P(\boldsymbol{z}; \boldsymbol{\theta})\right) \qquad (14)$$
$$= \min_g \log\left(1 + \exp\left(-\mathbb{E}_{\boldsymbol{z} \sim Q(\boldsymbol{z};\boldsymbol{\phi})} g(\boldsymbol{z}; \boldsymbol{\psi})\right)\right)$$
$$+ \log\left(1 + \exp\left(\mathbb{E}_{\boldsymbol{z} \sim P(\boldsymbol{z};\boldsymbol{\theta})} \exp(g(\boldsymbol{z}; \boldsymbol{\psi}))\right)\right).$$

Using the proposed KL approximation term, the full objective is to maximize

$$\mathcal{L}_{\text{CLVAE}}^{+} = \mathbb{E}_{\boldsymbol{z} \sim Q(\boldsymbol{z}|\boldsymbol{x},\boldsymbol{\phi})} \log P(\boldsymbol{x}|\boldsymbol{z}, \boldsymbol{\theta}) + \mathcal{L}_{\text{CL}}$$
$$- \log\left(1 + \exp\left(-\mathbb{E}_{\boldsymbol{z} \sim Q(\boldsymbol{z};\boldsymbol{\phi})} f(\boldsymbol{x}, \boldsymbol{z}; \boldsymbol{\psi})\right)\right)$$
$$- \log\left(1 + \exp\left(\mathbb{E}_{\boldsymbol{z} \sim P(\boldsymbol{z};\boldsymbol{\theta})} \exp(f(\boldsymbol{x}, \boldsymbol{z}; \boldsymbol{\psi}))\right)\right), \quad (15)$$
$$\mathcal{L}_{\text{CLVAE}_{\text{MI}}}^{+} = \mathbb{E}_{\boldsymbol{z} \sim Q(\boldsymbol{z}|\boldsymbol{x},\boldsymbol{\phi})} \log P(\boldsymbol{x}|\boldsymbol{z}, \boldsymbol{\theta}) + \mathcal{L}_{\text{CL}}$$
$$- \log\left(1 + \exp\left(-\mathbb{E}_{\boldsymbol{z} \sim Q(\boldsymbol{z};\boldsymbol{\phi})} g(\boldsymbol{z}; \boldsymbol{\psi})\right)\right)$$
$$- \log\left(1 + \exp\left(\mathbb{E}_{\boldsymbol{z} \sim P(\boldsymbol{z};\boldsymbol{\theta})} \exp(g(\boldsymbol{z}; \boldsymbol{\psi}))\right)\right). \quad (16)$$

**Training Algorithm** Algorithm 1 shows the training algorithm of our proposed method. We first sample a mini-batch of paired random Gaussian noise vectors $\boldsymbol{\xi}_i$ and $\boldsymbol{\xi}_i^+$. After obtaining a mini-batch of input sentences, we pass them through the LSTM encoder with dropout being enabled to produce the latent vectors $\boldsymbol{z}_{\boldsymbol{x},i}$ and $\boldsymbol{z}_{\boldsymbol{x},i}^+$. Then the contrastive loss defined in Eq 10 is calculated, and a paired prior vectors are sampled from $P(\boldsymbol{z}; \boldsymbol{\theta})$. $\boldsymbol{\psi}$ in $f(\boldsymbol{x}, \boldsymbol{z}, \boldsymbol{\psi})$ is updated according to Eq 18. Here we further consider to minimize the differences between the dual functions of the same input data by defining a squared loss $L_{\text{SQ}}$, which is given by

$$L_{\text{SQ}} = \sum_i \left(f(\boldsymbol{x}_i, \boldsymbol{z}_{\boldsymbol{x},i}; \boldsymbol{\psi}) - f(\boldsymbol{x}_i, \boldsymbol{z}_{\boldsymbol{x},i}^+; \boldsymbol{\psi})\right). \quad (17)$$

Given the same input and different $z$, the values of function $f$ are forced to reside in a small region. $\boldsymbol{\psi}$ is fixed afterwards. The encoder parameters $\boldsymbol{\phi}$ and decoder parameters $\boldsymbol{\theta}$ are updated according to Eq 19. If the dual function $g$ instead of $f$ is used, a similar loss as Eq 17 can be defined using $g$, and Eq 18 and Eq 19 can be changed accordingly.

# 4 EXPERIMENTS

We evaluate our proposed models on two tasks: 1) language modeling; 2) a synthetic clustering setting;

## 4.1 DATA AND SETTINGS

**Language Modeling** For language modeling, we consider three datasets, including the Penn Treebank [Marcus et al.,

**Input**: The training data set $D$ and the training epochs $T$;
**Model parameters**: $\boldsymbol{\theta}$, $\boldsymbol{\phi}$ and $\boldsymbol{\psi}$;
**while** $t < T$ **do**
  1. Sample a mini-batch of $\boldsymbol{\xi}_i \sim \mathcal{N}(\boldsymbol{\xi}), \boldsymbol{\xi}_i^+ \sim \mathcal{N}(\boldsymbol{\xi})$;
  2. Sample a mini-batch of input sentences $\boldsymbol{x}_i \sim \mathcal{D}$;
  3. Generate $\boldsymbol{z}_{\boldsymbol{x},i} = \text{ENC}(\boldsymbol{x}_i, \boldsymbol{\xi}_i; \boldsymbol{\phi})$ and
    $\boldsymbol{z}_{\boldsymbol{x},i}^+ = \text{ENC}(\boldsymbol{x}_i, \boldsymbol{\xi}_i^+; \boldsymbol{\phi})$;
  4. Calculate $\mathcal{L}_{\text{CL}}$ in Eq 10;
  5. Sample a mini-batch of $\boldsymbol{z}_i, \boldsymbol{z}_i^+ \sim P(\boldsymbol{z}; \boldsymbol{\theta})$;
  6. Update $\boldsymbol{\psi}$ in $f(\boldsymbol{x}, \boldsymbol{z}, \boldsymbol{\psi})$ to minimize
$$L_{\text{SQ}} + \log\left(1 + \exp(\sum_i -f(\boldsymbol{x}_i, \boldsymbol{z}_{\boldsymbol{x},i}; \boldsymbol{\psi}))\right)$$
$$+ \log\left(1 + \exp(\sum_i \exp(f(\boldsymbol{x}_i, \boldsymbol{z}_i; \boldsymbol{\psi})))\right)$$
$$+ \log\left(1 + \exp(\sum_i -f(\boldsymbol{x}_i, \boldsymbol{z}_{\boldsymbol{x},i}^+; \boldsymbol{\psi}))\right)$$
$$+ \log\left(1 + \exp(\sum_i \exp(f(\boldsymbol{x}_i, \boldsymbol{z}_i^+; \boldsymbol{\psi})))\right). \quad (18)$$
  7. Update parameters $\{\boldsymbol{\phi}, \boldsymbol{\theta}\}$ to minimize
$$- \sum_i \log P(\boldsymbol{x}_i|\boldsymbol{z}_i, \boldsymbol{\theta}) - \sum_i \log P(\boldsymbol{x}_i|\boldsymbol{z}_i^+, \boldsymbol{\theta}) - \mathcal{L}_{\text{CL}}$$
$$+ \log\left(1 + \exp\left(-\sum_i f(\boldsymbol{x}_i, \boldsymbol{z}_{\boldsymbol{x},i}; \boldsymbol{\psi}))\right)\right)$$
$$+ \log\left(1 + \exp\left(-\sum_i f(\boldsymbol{x}_i, \boldsymbol{z}_{\boldsymbol{x},i}^+; \boldsymbol{\psi}))\right)\right). \quad (19)$$
  8. $t \leftarrow t + 1$;
**Output**: $\boldsymbol{\theta}$, $\boldsymbol{\phi}$ and $\boldsymbol{\psi}$;
**Algorithm 1:** The training algorithm (a single step SGD) for contrastive learning over latent variables for text generation.

1993], Yahoo [Yang et al., 2017] and Yelp [He et al., 2019]. The Penn Treebank (PTB) is a common benchmark for language modeling, which consists of 42K sentences of varying lengths and 1 million words of Wall Street Journal material in 1989. Yahoo and Yelp are much larger datasets compared to the PTB. They both consist of 100K sentences and the average length of them are 78.7 and 96.0 words, respectively. Each dataset contains train, validation and test sets. The detailed data statistics are in Appendix A.1. We select the best model according to the performance on validation set and report the results on the test set using the corresponding best model.

**Synthetic Clustering Data** Following Fang et al. [2019], we design an experiment of synthetic clustering to show the learning dynamics of the induced latent variable. Given a random input category $k$, $k \in [0, K - 1]$ and $K$ is the number of categories, a 2-dimensional random Gaussian noise sampled from $\mathcal{N}(0, 1)$ is combined with the one-hot representation of the input category as the input variable $\mathbf{x}$. An encoder with multi-layered feedforward neural networks transforms $\mathbf{x}$ into a 2-dimensional latent vector $\mathbf{z}$. An decoder is trained to reconstruct $\mathbf{x}$ from $\mathbf{z}$. Suppose that the reconstructed output is $\mathbf{y}$ and $\mathbf{y} \in \mathbb{R}^K$, the binary cross-entropy loss is used, namely

Table 1: Language modeling on PTB.

| METHODS | -ELBO↓ | PPL↓ | KL↑ | MI↑ | AU↑ |
|---|---|---|---|---|---|
| VAE | 102.6 | 108.26 | 1.08 | 0.8 | 2 |
| $\beta_{0.5}$VAE | 104.5 | 117.92 | 7.50 | 3.1 | 5 |
| $Sa$VAE | 102.6 | 107.71 | 1.23 | 0.7 | 2 |
| $Cyc$VAE | 103.1 | 110.50 | 3.48 | 1.8 | 5 |
| $i$VAE | 87.6 | 54.46 | 6.32 | 3.5 | **32** |
| $i$VAE$_{MI}$ | 87.2 | 53.44 | 12.51 | 12.2 | **32** |
| **clVAE** | 82.7 | 43.51 | 8.37 | 5.55 | **32** |
| **clVAE**$_{MI}$ | 82.7 | 43.49 | **13.95** | **13.69** | **32** |
| **clVAE**$^{+}$ | 79.8 | 38.04 | 7.01 | 4.72 | **32** |
| **clVAE**$^{+}{}_{MI}$ | **77.7** | **34.61** | 9.94 | 9.58 | **32** |

Table 2: Language modeling on Yahoo and Yelp.

| METHODS | -ELBO↓ | PPL↓ | KL↑ | MI↑ | AU↑ |
|---|---|---|---|---|---|
| DATASET: Yahoo | | | | | |
| VAE | 328.6 | 61.21 | 0.0 | 0.0 | 0 |
| $\beta_{0.4}$VAE | 328.7 | 61.29 | 6.3 | 2.8 | 8 |
| $Sa$VAE | 327.2 | 60.15 | 5.2 | 2.7 | 10 |
| $Lag$VAE | 326.7 | 59.77 | 5.7 | 2.9 | 15 |
| $i$VAE | 309.5 | 48.22 | 8.0 | 4.4 | **32** |
| $i$VAE$_{MI}$ | 309.1 | 47.93 | **11.4** | **10.7** | **32** |
| **clVAE**$^{+}{}_{MI}$ | **303.3** | **44.61** | 8.6 | 7.7 | **32** |
| DATASET: Yelp | | | | | |
| VAE | 357.9 | 40.56 | 0.0 | 0.0 | 0 |
| $\beta_{0.4}$VAE | 358.2 | 40.69 | 4.2 | 2.0 | 4 |
| $Sa$VAE | 355.9 | 39.73 | 2.8 | 1.7 | 8 |
| $Lag$VAE | 355.9 | 39.73 | 3.8 | 2.4 | 11 |
| $i$VAE | 348.2 | 36.70 | 7.6 | 4.6 | **32** |
| $i$VAE$_{MI}$ | 348.7 | 36.88 | **11.6** | **11.0** | **32** |
| **clVAE**$^{+}{}_{MI}$ | **343.6** | **34.97** | 8.8 | 8.2 | **32** |

$L = -\mathbf{e}_k \log \sigma(\mathbf{y}_k) + \sum_{i \neq k}(1 - \mathbf{e}_i)\log(1 - \sigma(\mathbf{y}_i))$, where $k$ is the input category, $\mathbf{e}$ is the corresponding one-hot vector representation of the category $k$ and $\sigma$ is the sigmoid function $\sigma(x) = \frac{1}{1+\exp(-x)}$.

During training, we generate a batch of one-hot vectors by randomly choosing from $[1, 0, 0, 0]$, $[0, 1, 0, 0]$, $[0, 0, 1, 0]$ and $[0, 0, 0, 1]$ when $K = 4$. The batch size is 256 and we train the encoder-decoder 30K to 80K epochs until convergence. Ideally, the cluster of latent variable $\mathbf{z}$ in 2 dimensional space should exactly correspond to the input category after training. To understand the training behaviour of our proposed model, we visualize $\mathbf{z}$ in 2D space.

## 4.2 LANGUAGE MODELING RESULTS

**PTB Results** We compare our models with state-of-the-art VAE language models without using contrastive learning, including: 1) traditional VAE models with a KL-annealing strategy [Bowman et al., 2015]; 2) $\beta$VAE [Higgins et al., 2017], which controls the penalty on KL using a small hyper-parameter $\beta$; 3) $Sa$VAE [Kim et al., 2018], which is a semi-amortized VAE; 4) $Cyc$VAE [Fu et al., 2019], which anneals the KL term in a cyclical way; 5) $lag$VAE [He et al., 2019], which lags the update of decoder by aggressively updating encoder several times; 6) $i$VAE [Fang et al., 2019], which assumes an implicit distribution of latent variable as mentioned before; 7) $i$VAE$_{MI}$ is an enhanced version of $i$VAE by directly considering the mutual information between the input $\mathbf{x}$ and the latent variable $\mathbf{z}$. We denote our model as **clVAE** (Eq 11) and the circle loss enhanced version is named as **clVAE**$^{+}$ (Eq 15). **clVAE**$_{MI}$ (Eq 12) and **clVAE**$^{+}{}_{MI}$ (Eq 16) are their corresponding enhanced versions by mutual information.

We evaluate the models in terms of the quality of both the generation texts and the induced latent features. To measure the generation outputs, we use negative ELBO scores (-ELBO) and perplexity scores (PPL). The lower they are, the better the generated outputs are. Following Fang et al. [2019], we evaluate the quality of induced latent features in terms of $\mathsf{KL}(Q(\mathbf{z}|\mathbf{x}; \phi)||P(\mathbf{z}; \boldsymbol{\theta}))$, the mutual information

$I(\mathbf{x}, \mathbf{z})$ under the joint distribution of $Q(\mathbf{x}, \mathbf{z}; \phi)$, and the number of active units of $\mathbf{z}$, which is defined as $\text{AU}(\mathbf{z}) = \text{Cov}_{\mathbf{x}}(\mathbb{E}_{z \sim Q(\mathbf{z}|\mathbf{x}; \phi)}[z]) > 0.01$. Since all these terms cannot be solved in an analytical way, they are approximated by sample $\mathbf{z}$ 128 times from $Q(\mathbf{z}|\mathbf{x}; \phi)$. In general, higher KL, larger MI and AU values indicate the latent feature is more sufficiently used by the model.

Table 1 shows the main results on PTB test set. Our simplest model **clVAE** outperforms all the baselines, which shows that adding contrastive learning over latent variables greatly improves the model performance. **clVAE** reduces ELBO and PPL by 4.9 and 10.95 points compared to $i$VAE. The KL and MI values are increased by 2.05 points compared to $i$VAE. There results show that our model make better use of the latent vector compared to baselines, and produce better outputs. Similar conclusions hold for the models enhanced by mutual information, which suggests that contrastive learning can be complementary to mutual information estimation. Particularly, **clVAE**$_{MI}$ gives the best KL and MI values among all the models, demonstrating that it is a good choice to use the learned latent features. Using the proposed circle loss enhancement, the model performance improves with respect to ELBO and PPL, with 79.8 ELBO and 38.04 PPL, which are much better than **clVAE** (82.7 ELBO and 43.51 PPL). Among them, **clVAE**$^{+}{}_{MI}$ gives the best results, presenting 77.7 ELBO and 34.61 PPL. In terms of KL and MI, using the circle loss enhancement slightly hurts the sufficient use of latent variables compared to **clVAE**$_{MI}$. However, the KL and MI values are relative large, ranking the third among all the models. Therefore, considering both the generation quality and the effective use of latent space, the circle loss enhancement is useful.

**Yahoo and Yelp Results** We use the best configuration model of **clVAE**$^{+}{}_{MI}$ to conduct experiments on Yahoo and Yelp. Table 2 shows the results. On Yahoo, our model outper-

Table 3: Contrastive learning using **h** or **z**.

| WHERE | METHODS | -ELBO↓ | PPL↓ | KL↑ | MI↑ | AU↑ |
|---|---|---|---|---|---|---|
| **h** | **cl**VAE | 83.0 | 44.10 | 7.51 | 4.92 | **32** |
| | **cl**VAE$_{MI}$ | 86.7 | 52.13 | 13.35 | 13.09 | **32** |
| **z** | **cl**VAE | **82.7** | 43.51 | 8.37 | 5.55 | **32** |
| | **cl**VAE$_{MI}$ | **82.7** | **43.49** | **13.95** | **13.69** | **32** |

Table 4: Forward and reverse PPL on PTB test set.

| MODEL | FORWARD↓ | REVERSE↓ |
|---|---|---|
| VAE | 18,494 | 10,149 |
| $Cyc$VAE | 3,390 | 5,587 |
| AE | 672 | 2,589 |
| $\beta_0$VAE | 625 | 1,897 |
| $\beta_{0.5}$VAE | 939 | 4,078 |
| $Sa$VAE | 341 | 10,651 |
| $i$VAE | **116** | 1,520 |
| $i$VAE$_{MI}$ | 134 | 1,137 |
| **cl**VAE$^+{}_{MI}$ | 120 | **1,047** |

form $i$VAE$_{MI}$ in terms of both ELBO and PPL. For ELBO, our model outperform $i$VAE$_{MI}$ by 5.8 points. For PPL, our model gives 44.61 points, reducing the PPL of $i$VAE$_{MI}$ by 3.32 points. For KL and MI values, our model are better than $i$VAE and comparable to $i$VAE$_{MI}$. Similar observations can be found on Yelp dataset. **cl**VAE$^+{}_{MI}$ reduces ELBO and PPL by 5.1 and 1.91 points compared to $i$VAE$_{MI}$. On Yelp, for KL and MI values are 8.8 and 8.2, respectively, ranking the second. Since both Yahoo and Yelp are large corpora, it is challenging to greatly reduce the PPL and ELBO scores while maintaining high KL and MI values at the same time. Our model is empirically well balanced.

## 4.3 ANALYSIS

**The effect of contrastive learning for latent variable** Table 3 shows the comparison results of doing contrastive learning using the encoder output **h** and the latent variable **z** on PTB test set, respectively. For simplicity, we use two basic models: namely **cl**VAE and **cl**VAE$_{MI}$. As shown in Table 1, when the same models are used, doing contrastive learning over **z** instead of **h** gives better results. For example, when using the **cl**VAE$_{MI}$ model, the model based on **h** gives 52.13 PPL, while the model based on **z** obtains 43.49 PPL, which is much lower than the model based on **h**. Only in terms of ELBO and PPL values, the basic model **cl**VAE over **z** even performs better than the mutual information enhanced **cl**VAE$_{MI}$ models over **h**. Besides, when doing contrastive learning on **h**, the performance gap between **cl**VAE and **cl**VAE$_{MI}$ are large, whereas the corresponding gap is shortened when doing contrastive learning on **z**. These results suggest that it is necessary to do contrastive learning over latent variables, producing better results compared to directly doing contrastive learning on the encoder hidden output.

**The effect on the decoder** To evaluate whether the decoder

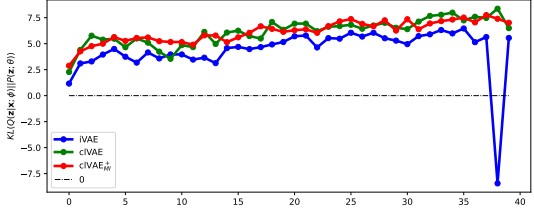

(a) Estimated $KL(Q(\mathbf{z}|\mathbf{x},\phi)||P(\mathbf{z},\boldsymbol{\theta}))$ w.r.t training epoch.

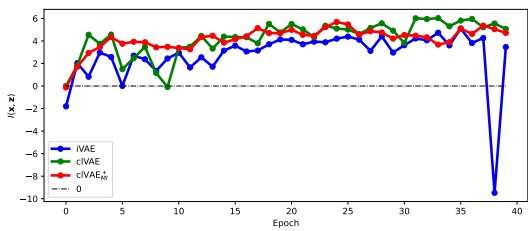

(b) Mutual information $I(\mathbf{x}, \mathbf{z})$ w.r.t training epoch.

Figure 3: KL and MI w.r.t the training epoch.

is being improved by making better use of latent features, we sample latent variables from the prior distribution $P(\mathbf{z})$ and ask the decoder to generate outputs based on the sampled latent codes, following Fang et al. [2019] and Kim et al. [2017]. The generated text is evaluated by KenLM [Heafield et al., 2013] using two metrics: forward PPL and reverse PPL. The forward PPL evaluates the generated texts based on a language model trained on the PTB-train corpus. Lower forward PPL indicates the generated texts are more fluent. The reverse PPL evaluates the PTB corpus based on a language model trained on the generated texts. Lower reverse PPL means the generated texts are better representative of the PTB corpus. For the underlying language model, 5-gram KenLM is used.

Table 4 shows the forward PPL and reverse PPL on the PTB. For reverse PPL, our **cl**VAE$^+$performs better than all the baselines and **cl**VAE$_{MI}$ gives the best values, which shows that our models can better represent the true data distribution. For forward PPL, our model performs better than most of baselines models including $Sa$VAE and $\beta_{0.5}$VAE. iVAE gives the best forward PPL scores (116) since it imposes less constraints on the latent variable and thus benefits from a smoother space. Our model give 120 forward PPL scores, which is comparable to iVAE and acceptable by sacrificing a little fluency, but it can generate more diverse outputs and make text appear more human like.

**The negativity of approximate KL and MI terms** Section 3.3 mentioned that the approximated KL and MI term can become negative during training iVAE. Empirically, we demonstrate this problem by using the PTB language modeling task. Figure 3a and Figure 3b show how the KL term and the mutual information term change with the training epoch of the three models, including iVAE, **cl**VAE and **cl**VAE$^+{}_{MI}$, respectively. As shown in Figure 3a, the approximate KL

Table 5: Comparisons of iVAE and **cl**VAE$^+$ on toy data.

(a) $K = 4$

| Model | 0k | 5k | 15k | 30k | Final |
|---|---|---|---|---|---|
| VAE | | | | | |
| iVAE | | | | | |
| **(Ours)** | | | | | |

(b) $K > 4$

| Model | 0k | 5k | 30k | 50k | 80k |
|---|---|---|---|---|---|
| iVAE ($K = 8$) | | | | | |
| **Ours** ($K = 8$) | | | | | |
| **Ours** ($K = 16$) | | | | | |

term of iVAE suddenly becomes a large negative number at the 39th epoch. From Figure 3b, we can observe that the approximate MI term at 0th epoch, 2nd epoch and the 39th epoch are all negative. This observation shows that the negativity problem of approximate KL and MI terms do exist in iVAE. Adding the contrastive learning module alleviates this problem to some extent. The KL and MI values of **cl**VAE become more stable than iVAE. However, the KL value of **cl**VAE dramatically decreases at the 10th epoch. In contrast, no such phenomenons exist in our **cl**VAE$^+{}_{\text{MI}}$ model based on the empirical observations. Its KL and MI values gradually increase as the training goes on in general. These evidences suggest that our proposed circle loss enhancement can learn a more robust model in term of the stability of the approximated KL and MI values.

## 4.4 RESULTS ON SYNTHETIC DATA

**$K = 4$** Table 5a shows the visualizations of the learned 2D latent vector of VAE, iVAE and our model with different training epochs. When epoch $= 0K$, the distribution of the induced latent variable of VAE is basically a normal distribution and the points are mixed together. Since both iVAE and our model represents the latent variable using an implicit distribution, there are no clues about the distribution of the latent variable in the beginning. After training with $5K$ epochs, we observe that VAE can make an initial guess concerning the corresponding cluster of the in-

duced latent variable, iVAE is still struggling to cluster these points, whereas our model can achieve a clear separation between clusters, and capture well the latent distribution of the data. As the training continues to 15K and 30K epochs, VAE shapes the data distribution as a sandwich, as shown in Table2, the green and blue points are widely mixed together and the data points within the same cluster are wide spread. Meanwhile, iVAE still could not provide a nearly clear separation between clusters. However, our model starts to shorten the distance among points within the same cluster and enlarge the distance between different clusters. As a fact, our model only needs 3.5K epochs to converge while both VAE and iVAE take 80K epochs to converge. After 80k epochs, iVAE divides the 2D space into the corresponding clusters successfully, and it can provide a better data distribution compared to VAE. However, the data points with the same cluster given by iVAE are dispersed. As shown in Table 5a, our model converges much faster than VAE and iVAE. It also gives a more compact and coherent representation, making the intra-class distance smaller and the inter-class distance larger compared to VAE and iVAE. This shows that adding contrastive learning over latent spaces can better regularize the distributions of latent variables.

**$K > 4$** When the number of category becomes large, the difficulty of representing data points in the 2D latent variable also increases. Table 5b shows the comparisons of converging trend between iVAE and our model when the input category is larger than $K = 4$. When $K = 8$, iVAE fails to converge even using $80K$ training epochs, while our model obtains a decent separation boundaries using only 5K epochs. After training $80K$ epochs, our model successfully cluster the input data into the corresponding category. To show the capability of our method, we further set $K = 16$, making it more challenging. In this setting, the class boundaries appear after 5K epochs, but not so clear as those when $K = 8$ using the same training budget. Separation between clusters gets refined as the iterations go on. With 80K training epochs, our models can manage to reconstruct the input categories from the 2D latent variable. This not only shows the discriminatory nature of the learnt representations, but also the speed with which becomes a potential advantages of combining contrastive learning with latent variables, and the capacity of handling more complex situations compared to the traditional latent variable models.

## 4.5 SENTENCE INTERPOLATION

Table 6 shows the sentence interpolation [Bowman et al., 2015] results of two example sentences. Similar to VAE, our model can generate sentences by interpolating the latent semantic vectors. Given two sentences $\mathbf{x}_1$ and $\mathbf{x_2}$, we generate vectors $\mathbf{z}_1$ and $\mathbf{z}_2$ by averaging samples from $Q_\phi(\mathbf{z}|\mathbf{x})$. Given $\mathbf{z}_1$ and $\mathbf{z}_2$, we generate a new latent vector $\mathbf{z} = \lambda * \mathbf{z}_1 + (1 - \lambda) * \mathbf{z}_2$ by interpolating the sentence

Table 6: Interpolation of latent representation.

| | |
|---|---|
| $\lambda = 0$ | there was \$ N billion and more interest income |
| $\lambda = 0.1$ | there was \$ N billion and more interest in 30-year |
| $\lambda = 0.2$ | there was \$ N billion and more than \$ N billion |
| $\lambda = 0.3$ | there was \$ N million more than two days |
| $\lambda = 0.4$ | there was \$ N million more than in chicago |
| $\lambda = 0.5$ | we had \$ N million in the stock market |
| $\lambda = 0.6$ | we had \$ N million in the latest period |
| $\lambda = 0.7$ | we had \$ N million in the latest period |
| $\lambda = 0.8$ | we 'll N years in its latest day |
| $\lambda = 0.9$ | i went in N with a few months ago |
| $\lambda = 1$ | i went in N with a few months in N |

semantics of $\mathbf{z}_1$ and $\mathbf{z}_2$. Then $\mathbf{z}$ is used by the decoder to produce a sentence with mixed semantics. $\lambda$ is varied from 0 to 1 with a step size of 0.1. As shown in Table 6, our model can generate sentences by smooth considering the semantics from the two input sentences.

### 4.6 EFFECT OF DIFFERENT DROPOUT RATES

Figure 4: PPL and ELBO values w.r.t dropout rate.

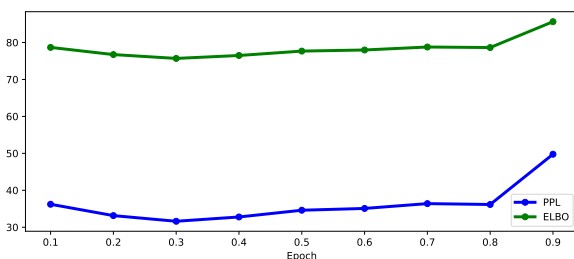

We use the PTB to study the effect of different dropout rates by varying the dropout ratio from 0.1 to 0.9 with a step size 0.1. Figure 4 shows the PPL and ELBO values of different dropout rates. We find that $dropout = 0.3$ gives the optimal performance. Empirically, increasing the dropout rates from 0.1 to 0.3, the performances become better. When the dropout ratio is in range [0.3, 0.6], the performances are stable. Using a large dropout rate ($dropout >= 0.7$), the performances degrade.

### 4.7 DIFFERENT DATA AUGMENTATION METHODS

The selection of positive and negative samples are very important for the success of contrastive learning. In this paper, we follow Gao et al. [2021] and use dropout as the minimum data augmentation. Applying dropout twice to a recurrent encoder for the same input sentence can lead to two different (but still semantically related) posterior distributions. The latent samples which represent the two implicit posterior

Table 7: Comparisons with different augmentation methods.

| AUGMENTATION | -ELBO↓ | PPL↓ | KL↑ | MI↑ | AU↑ |
|---|---|---|---|---|---|
| SWAP | 79.1 | 36.96 | 9.21 | 8.95 | 32 |
| DROPOUT | **77.7** | **34.61** | **9.94** | **9.58** | 32 |

distributions can be sufficiently different, therefore being useful for contrastive learning for variational auto-encoders. We additionally perform experiments by using random swap to create negative samples. Given an input sentence, we flip a coin with a probability 0.1 to decide whether to swap the positions of the $i$-th token and the (i+1)-th token, from $i = 1$ to $i = n - 1$. In this way, the word order is perturbed. We compare the dropout-based augmentation with swap-based augmentation. Table 7 shows that dropout is a better augmentation method for the proposed model than swap-based method. Considering more different data augmentations will be considered in future work.

## 5 CONCLUSION

We proposed contrastive learning over latent variables in variational autoencoders (VAE), designing a circle loss enhancement to solve the approximation problem of the KL term used in a state-of-the-art VAE models. Experiments showed that combining contrastive learning with latent variable models can both improve the generation quality and make sufficient use of latent space. In future, we intend to expand our framework to graph and image data.

### Author Contributions

Zhiyang Teng proposed the main idea, wrote the code and the paper, and performed main experiments. Chenhua Chen performed auxliary experiments and polished the paper. Yan Zhang participated in the discussion of the method of contrastive learning and wrote part of the related work. Yue Zhang provided insights to claim the main idea and proof-read the writings.

### Acknowledgements

Yue Zhang is the corresponding author. This work is sponsored by CCAI-Huawei MindSpore Open Fund and the Zhejiang Province Key Project 2022SDXHDX0003. We thank all the reviewers for their valuable comments and suggestions. We gratefully acknowledge the support of MindSpore, CANN (Compute Architecture for Neural Networks) and Ascend AI Processor used for this research.

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
