# OpenReview forum: "Contrastive Latent Variable Models for Neural Text Generation"
_auai.org/UAI/2022/Conference — UAI 2022 Poster_

### Official Review · Reviewer_uEWV · 2022-04-08

**Q2(1) Originality/Novelty:** 3
**Q2(2) Significance/Impact:** 3
**Q2(3) Correctness/Technical Quality:** 4
**Q2(6) Clarity Of Writing:** 4
**Q6 Overall Score:** 7
**Q8 Confidence In Your Score:** 4

**Q1 Summary And Contributions:**

There has been a long line of work on using VAEs for text, and Vanilla VAEs suffer from a number of issues such as posterior collapse.

The paper proposes introducing a contrastive loss term in the latent space. A latent space defines a many-to-one mapping from latent codes to decoded sequences. They use a triplet loss to organize this latent space so that two encodings of two 'views' of the same input sequence are closer than the encoding of an alternative sequence.


**Q2 Assessment Of The Paper:**

More detailed information regarding each of these aspects is given below:

**Q2(4) Quality Of Experiments (Optional):**

3: Good: The experimental evaluation is adequate, and the results convincingly support the main claims.

**Q2(5) Reproducibility:**

3: Good: Key resources (e.g., proofs, code, data) are available and key details (e.g., proofs, experimental setup) are sufficiently well-described for competent researchers to confidently reproduce the main results.

**Q3 Main Strengths:**

The main methodological contribution of the paper is compelling and will be well received by the community.

A number of innovations have been introduced to avoid issues of VAEs for text. This paper is carefully situated in this line of work, and is systematic with ablations to understand the importance of their contribution.

The experimental setup is good in that there are good ablations and strong baselines. However, they are mostly 'toy' problems (e.g. using PTB for language modeling).

**Q4 Main Weakness:**

The improvement over SOTA is small, and sometimes zero, for the method. However, In part this is because the paper compares against really strong baselines that use a number of cutting-edge techniques.



**Q5 Detailed Comments To The Authors:**

Can you please clarify this. I didn't understand:
"iVAE gives the best forward PPL scores (116) since it imposes
less constraints on the latent variable and thus benefits from
a smoother space."

I didn't understand why you needed to use dropout to define two different views of the same input sequence. Wouldn't it be sufficient to use two different draws from the posterior distribution for a given sequence?

I didn't follow your algorithm box. What does 'update parameters to minimize' mean? Do you run multiple steps of SGD? How do you tune this?


**Q7 Justification For Your Score:**

The proposed modeling technique is a good idea that could be used in other contexts. The paper does an exemplary job of benchmarking against strong baselines and providing systematic ablations.



**Q9 Complying With Reviewing Instructions:**

1: Yes.

---

### Official Review · Reviewer_yTV1 · 2022-04-14

**Q2(1) Originality/Novelty:** 2
**Q2(2) Significance/Impact:** 2
**Q2(3) Correctness/Technical Quality:** 2
**Q2(6) Clarity Of Writing:** 2
**Q6 Overall Score:** 6
**Q8 Confidence In Your Score:** 3

**Q1 Summary And Contributions:**

The authors introduced constrastic learning to VAE-based text representation and generation. Then they use several widely studied NLP data sets to demonstrate that the latent state representation is indeed more "structured".

**Q2 Assessment Of The Paper:**

More detailed information regarding each of these aspects is given below:

**Q2(4) Quality Of Experiments (Optional):**

2: Fair: The experimental evaluation is weak: important baselines are missing, or the results do not adequately support the main claims.

**Q2(5) Reproducibility:**

2: Fair: Key resources (e.g., proofs, code, data) are unavailable but key details (e.g., proof sketches, experimental setup) are sufficiently well-described for an expert to confidently reproduce the main results.

**Q3 Main Strengths:**

The introduction of constrastic learning to VAE-based text representation.

**Q4 Main Weakness:**

- Lack of qualitative evaluation of text representation and generation by human experts.
- Lack of clarity on how negative samples are constructed.
- It is not clear how text generation is controllable with the latent representation.
- Overall, the work is incremental.

**Q5 Detailed Comments To The Authors:**

How negative samples are constructed, and how to achieve controllable text generation?


**Q7 Justification For Your Score:**

Overall, VAE-based text generation and representation is worth studying. But the work hasn't addressed the issues with VAE-based approach.

**Q9 Complying With Reviewing Instructions:**

1: Yes.

---

### Official Review · Reviewer_HK5t · 2022-04-15

**Q2(1) Originality/Novelty:** 3
**Q2(2) Significance/Impact:** 3
**Q2(3) Correctness/Technical Quality:** 3
**Q2(6) Clarity Of Writing:** 3
**Q6 Overall Score:** 5
**Q8 Confidence In Your Score:** 4

**Q1 Summary And Contributions:**

This paper presents a contrastive learning method in variational autoencoders (VAE), enforcing semantic faithfulness in text generation. The authors use two latent vectors corresponding to the same input as positive pairs and the two latent vectors belonging to different inputs as negative pairs. The experiments on a synthetic dataset and three language modeling benchmarks show the effectiveness of the proposed method.

**Q2 Assessment Of The Paper:**

More detailed information regarding each of these aspects is given below:

**Q2(4) Quality Of Experiments (Optional):**

2: Fair: The experimental evaluation is weak: important baselines are missing, or the results do not adequately support the main claims.

**Q2(5) Reproducibility:**

3: Good: Key resources (e.g., proofs, code, data) are available and key details (e.g., proofs, experimental setup) are sufficiently well-described for competent researchers to confidently reproduce the main results.

**Q3 Main Strengths:**

S1: It is interesting to combine variational autoencoders with contrastive learning.
S2: This paper is clearly written and easy to follow.
S3: The experiments and analyses show quite positive results.

**Q4 Main Weakness:**

W1: The success of contrastive learning heavily depends on the selection of positive and negative samples. The settings of the positive and negative samples in this paper seems trivial. It will be nice to see different methods for positive and negative sample selection. In this paper, it seems that it is very easy to distinguish positive samples from negative ones.

W2: The title of this paper is for neural text generation. However, this paper only conducts experiments on synthetic dataset and language modeling. Typical test generations tasks, such as summarization and data2text, are not investigated. It makes the proposed method less convincing on the generation ability to other generation tasks.

**Q5 Detailed Comments To The Authors:**

This paper presents a contrastive learning method in variational autoencoders (VAE), enforcing semantic faithfulness in text generation. The authors use two latent vectors corresponding to the same input as positive pairs and the two latent vectors belonging to different inputs as negative pairs. The experiments on a synthetic dataset and three language modeling benchmarks show the effectiveness of the proposed method.

Strengths:
S1: It is interesting to combine variational autoencoders with contrastive learning.
S2: This paper is clearly written and easy to follow.
S3: The experiments and analyses show quite positive results.

Weaknesses:
W1: The success of contrastive learning heavily depends on the selection of positive and negative samples. The settings of the positive and negative samples in this paper seems trivial. It will be nice to see different methods for positive and negative sample selection. In this paper, it seems that it is very easy to distinguish positive samples from negative ones.

W2: The title of this paper is for neural text generation. However, this paper only conducts experiments on synthetic dataset and language modeling. Typical test generations tasks, such as summarization and data2text, are not investigated. It makes the proposed method less convincing on the generation ability to other generation tasks.

**Q7 Justification For Your Score:**

Q4

**Q9 Complying With Reviewing Instructions:**

1: Yes.

---

### Official Review · Reviewer_mkVp · 2022-04-16

**Q2(1) Originality/Novelty:** 3
**Q2(2) Significance/Impact:** 3
**Q2(3) Correctness/Technical Quality:** 3
**Q2(6) Clarity Of Writing:** 3
**Q6 Overall Score:** 7
**Q8 Confidence In Your Score:** 3

**Q1 Summary And Contributions:**

This paper proposed a latent variable model cIVAE for text generation. Based on the IVAE, the model integrates contrastive learning in the latent space to learn high-level semantics of latent variables, and uses the CircleLoss to strengthen the model to ensure the non-negativity of KL divergence. Experiments on various text generation benchmarks demonstrate the effectiveness of the model.


**Q2 Assessment Of The Paper:**

More detailed information regarding each of these aspects is given below:

**Q2(4) Quality Of Experiments (Optional):**

3: Good: The experimental evaluation is adequate, and the results convincingly support the main claims.

**Q2(5) Reproducibility:**

3: Good: Key resources (e.g., proofs, code, data) are available and key details (e.g., proofs, experimental setup) are sufficiently well-described for competent researchers to confidently reproduce the main results.

**Q3 Main Strengths:**

1.The paper proposed a simple but effective generation model.
2.The proposed model applies contrastive learning to learn rich semantics in the latent space.
3.The proposed model greatly reduces the PPL and ELBO scores on large datasets.

**Q4 Main Weakness:**

1.Figure 1 is not obvious that the positive and negative samples are not marked clearly; what does the red arrow point to? And the figure does not reflect the different dropouts for data augmentation mentioned in the text.
2.The expression of Equation 8 is wrong.
3.Spelling and grammar check is required. For example, Page 2 line 17 leanring --> learning.
4.The following paper should be added to the related work section:
Lee Seanie, Lee Bok Dong and Hwang Ju Sung. Contrastive Learning with Adversarial Perturbations for Conditional Text Generation. EMNLP, 2021.

**Q5 Detailed Comments To The Authors:**

1.The model first enhances the input sentence to obtain positive and negative samples, and then gets the corresponding posterior latent variables through sampling, so in essence, is it still using contrastive learning in the input space rather than the latent space?
2.The proposed model involves a lot of loss, how to find the balanced parameters between them?

**Q7 Justification For Your Score:**

I like the proposed model because it is not complicated and seems effective in the experiment.
The contributions are:
1.Using contrastive learning in latent space to generate text.
2.Alleviating the KL divergence collapse problem through contrastive learning.
3.Achieving the best results on both the ELBO and PPL on the three datasets.



**Q9 Complying With Reviewing Instructions:**

1: Yes.

---

### Decision · Program_Chairs · 2022-05-15

**Decision:**

Accept (Poster)

**Comment:**

Meta Review: The paper adapts contrastive losses to implicit VAEs and mutual information regularized implicit VAEs. Along the way, the paper tried to improve the KL estimates derived by the variational form of the KL divergence by encouraging non-negative estimates. The paper carried out an extensive set of comparisons to other types of VAEs and carried out more experiments including a qualitative evaluation in their response to the reviewers. Overall, the approach is well motivated, and the reviewers are all supportive. This paper is a clear accept.